# Revealing the Mechanism of the Bias Temperature Instability Effect of p-GaN Gate HEMTs by Time-Dependent Gate Breakdown Stress and Fast Sweeping Characterization

**DOI:** 10.3390/mi14051042

**Published:** 2023-05-12

**Authors:** Xiangdong Li, Meng Wang, Jincheng Zhang, Rui Gao, Hongyue Wang, Weitao Yang, Jiahui Yuan, Shuzhen You, Jingjing Chang, Zhihong Liu, Yue Hao

**Affiliations:** 1Guangzhou Wide Bandgap Semiconductor Innovation Center, Guangzhou Institute of Technology, Xidian University, Guangzhou 510555, China; mengm10612183@163.com (M.W.); yangweitao@xidian.edu.cn (W.Y.); yuanjh1126@163.com (J.Y.); youshuzhen@xidian.edu.cn (S.Y.); jjingchang@xidian.edu.cn (J.C.); zhliu@xidian.edu.cn (Z.L.); yhao@xidian.edu.cn (Y.H.); 2Key Laboratory of Wide Bandgap Semiconductor Materials and Devices, School of Microelectronics, Xidian University, Xi’an 710071, China; 3China Electronic Product Reliability and Environmental Testing Research Institute, Guangzhou 511370, China; r.gao90@ceprei.com (R.G.); wanghongyue@pku.edu.cn (H.W.)

**Keywords:** p-GaN gate HEMTs, TDGB, BTI, charge

## Abstract

The bias temperature instability (BTI) effect of p-GaN gate high-electron-mobility transistors (HEMTs) is a serious problem for reliability. To uncover the essential cause of this effect, in this paper, we precisely monitored the shifting process of the threshold voltage (V_TH_) of HEMTs under BTI stress by fast sweeping characterizations. The HEMTs without time-dependent gate breakdown (TDGB) stress featured a high V_TH_ shift of 0.62 V. In contrast, the HEMT that underwent 424 s of TDGB stress clearly saw a limited V_TH_ shift of 0.16 V. The mechanism is that the TDGB stress can induce a Schottky barrier lowering effect on the metal/p-GaN junction, thus boosting the hole injection from the gate metal to the p-GaN layer. This hole injection eventually improves the V_TH_ stability by replenishing the holes lost under BTI stress. It is the first time that we experimentally proved that the BTI effect of p-GaN gate HEMTs was directly dominated by the gate Schottky barrier that impeded the hole supply to the p-GaN layer.

## 1. Introduction

Gallium nitride high-electron-mobility transistors (GaN HEMTs) have been widely used in both radio frequency and power electronics [1,2,3], thanks to the superior performance in switching speed and switching losses [4,5,6]. For this new technique, the bias temperature instability (BTI) effect is one of the most crucial reliability problems that often take place, especially when the GaN HEMTs are operating under positive gate voltage bias [7,8,9,10,11,12,13,14,15,16,17,18,19]. To achieve an enhancement-mode operation, two gate architectures, including p-GaN gate HEMTs with a Schottky gate terminal and gate injection transistors (GIT) with an Ohmic gate terminal [5], have been adopted for mass production.

The Schottky gate contact highlights a low gate leakage current and a high threshold voltage (V_TH_), however, suffering a severe BTI effect with the V_TH_ shifting more than 1 V. Gate-bias-induced V_TH_ instabilities of the p-GaN gate HEMTs with a Schottky gate have been frequently discussed. One possible mechanism is the hole emission from the p-GaN inducing a hole deficiency in the p-GaN layer [18,20]. Canato et al. proposed that the electron trapping and hole trapping in the AlGaN barrier could be the cause of V_TH_ positive and negative shifts, separately [21]. Yang et al. believed the trapping could also happen in the metal/p-GaN Schottky depletion region [15]. Recently, Tang et al. found that the forward gate bias could endanger the metal/p-GaN Schottky junction by observing the electroluminescence emission [22]. Furthermore, the ultraviolet luminescence generated in the experiment was proven to mainly come from the p-GaN and the GaN channel due to the electron-hole radiative recombination. In view of the above observations, we infer that there must be a strong link between the hole supply and the metal/p-GaN Schottky barrier, which can significantly impact the BTI performance of the p-GaN gate HEMTs.

In this work, time-dependent gate breakdown (TDGB) measurements were first conducted on the Schottky-type p-GaN gate HEMTs to induce the Schottky barrier lowering. During the TDGB measurements, BTI measurements were periodically applied to monitor the V_TH_ of the device. Afterward, a technology computer-aided design (TCAD) simulation was conducted to simulate the hole current on the gate with different work functions. In the end, a physical model was proposed to explain the mechanism of the BTI effect of the p-GaN gate HEMTs.

## 2. Devices and Characterization

### 2.1. Devices

A schematic structure of the commercial p-GaN gate HEMTs is shown in Figure 1a. The gate length L_G_ is 0.5 μm, the gate width W_G_ is 155 μm, and the gate-to-drain distance L_GD_ is 3 μm. The stack of gate metal/p-GaN/AlGaN/GaN consists of a Schottky diode (metal/p-GaN) in series with a p-i-n diode (p-GaN/AlGaN/GaN) as shown in Figure 1b. The micrograph of the measured device is shown in Figure 1c. The epitaxy layer comprises a 4 μm thick AlGaN buffer layer, an unintentionally doped 200 nm GaN channel layer, a 15 nm Al_0.2_Ga_0.8_N barrier layer, and a 70 nm p-GaN layer doped with a Mg doping concentration of 3 × 10^19^ cm^−3^.

The static output and transfer curves of the devices were characterized by the Keysight B1500A semiconductor device analyzer, as shown in Figure 2a,b. The V_TH_ is 1.8 V, defined at I_D_ = 1 μA.

### 2.2. BTI Characterization

Traditional BTI measurements are normally based on the Measure-Stress-Measure (MSM) technique, which suffers a long relaxation time that could induce a totally fake shift value. Recently, the BTI measurements of the p-GaN gate HEMTs were successfully conducted by an Aglient B1530A waveform generator/fast measurement unit (WGFMU) with a relaxation time as short as 50 ns. The B1530A WGFMU is a new type of measurement unit that integrates arbitrary linear waveform generation (ALWG) capability with high-speed IV measurement, which can generate not only DC, but also various AC waveforms with 10 ns programmable resolution. This measurement sequence is named the extended Measure-Stress-Measure (eMSM) technique [23,24,25]. In this work, we also adopted this new technique to characterize the p-GaN gate HEMTs in order to gain insights into the cause of the BTI mechanism. As shown in Figure 3a, during the BTI measurements, the stress was periodically interrupted to measure I_D_-V_G_ curves by sweeping the V_GS_ from 0 to 3 V in 3 μs. During the BTI test, V_DS_ was kept at a low value of 0.2 V to avoid new stress on the device, and the source was grounded. Thanks to the negligible relaxation time, the BTI behavior can be precisely recorded by the fast I_D_-V_G_ sweep. To monitor the recovery process, a similar eMSM sequence was applied except for the V_GS_ = 0 V, and this process was also periodically interrupted by I_D_-V_G_ sweepings.

## 3. Results

### 3.1. BTI of the p-GaN Gate HEMTs

Figure 3a shows the I_D_-V_G_ curves of the device during the BTI eMSM measurements. The inset of Figure 3a is the plot of the same I_D_-V_G_ curves with a logarithmic scale on the y-axis. The V_TH_ sees a monotonous increase with the stress time t_stress_, consistent with our previous observation [25]. Figure 3b summarizes the evolution of the V_TH_ under various gate stress voltages, and the higher stress voltage induces a more pronounced shift. Afterward, the recovery process was characterized. As shown in Figure 4a,b, after tens of seconds, the V_TH_ fully recovers, indicating that the stress process does not generate new defects.

The root cause of the BTI effect of the p-GaN gate HEMTs is quite complicated, which is probably a mixed effect of trapping and charging. Kevin Chen et al. have pointed out that two main trap states exist in the p-GaN stack, which can aggravate the BTI effect [15]. More other works indicate that the charging effect should be more important [20,26]. For p-GaN gate HEMTs, the charging effect should be seriously considered because the p-GaN is floating. In detail, the potential barrier by the metal/p-GaN Schottky junction blocks the transport of holes from the gate to the p-GaN layer [18]. If this assumption holds, we can suppress the BTI effect by lowering the potential barrier.

### 3.2. Suppressing the BTI Effect by TDGB Stress

To lower the potential barrier of the metal/p-GaN Schottky junction, TDGB stress was applied to the p-GaN gate HEMTs. The experimental flow is demonstrated in Figure 5. The fresh BTI eMSM measurement was first conducted on the device to monitor the BTI behavior of the fresh device. Afterward, the device was subjected to TDGB stress, which was periodically interrupted by the BTI eMSM measurements to record the BTI performance of the device. Meanwhile, the I_G_-V_G_ was also measured during the interruption.

During the stress, the source and drain terminals were grounded, and the gate was positively biased at a constant voltage of 10 V. Figure 6a depicts the gate leakage versus the stress time. The stress was also periodically interrupted by BTI eMSM measurements to monitor the V_TH_ during the stress, as shown in Figure 6b. During the TDGB stress, the gate stress was mostly applied on the metal/p-GaN Schottky junction instead of the p-GaN/AlGaN/GaN p-i-n diode, because the former junction is reversely biased and the latter one is positively biased. In this way, the TDGB stress can introduce some damage to the Schottky junction, thus lowering the Schottky potential barrier for the holes.

As shown in Figure 6b, the fresh device without any TDGB stress demonstrates a serious V_TH_ shift by 0.62 V. However, this shift quickly jumps to 0.32 V for the second BTI measurement, only after 10 s of TDGB stress. After the cumulative stress time of 424 s, the V_TH_ shift decreases to a pretty low value of 0.16. From Figure 6b, we can clearly see that the V_TH_ surges during the initial 1 ms BTI stress. Once the BTI stress is applied on the gate, the gate voltage is mainly divided by the Schottky junction and the p-i-n junction, by capacitive coupling. The GaN is thus easily turned on at a relatively low gate voltage. As the stress time increases, part of the holes in the p-GaN surmount the p-GaN/AlGaN barrier and emit to the GaN channel. Therefore, the p-GaN is negatively charged, and the V_TH_ gradually increases. In the second phase from 1 ms to 30 ms, the V_TH_ sees a drop, probably due to the hole trapping in the AlGaN barrier layer. In the third phase from 30 ms, the V_TH_ becomes stable, probably because of the saturation of hole loss and hole trapping. The root cause of the BTI effect could partially include the trapping effect, which is anyhow unable to be ignored.

From Figure 7a, we can see that the drain current increases in the saturation region of the output curves, indicating a negative V_TH_ shift of the HEMT after the 424 s stress, consistent with the V_TH_ evolution in Figure 6b. Figure 7b directly demonstrates this shift, which is probably induced by the hole traps generated by the TDGB stress.

During the TDGB stress, the gate leakage was also monitored as shown in Figure 8a. The fresh forward I_GSS_ is relatively low. After cycles of TDGB stresses, the forward I_GSS_ gradually degraded. In contrast, the reverse I_GSS_ is more stable. This contrast directly proves that the degradation part is indeed the metal/p-GaN Schottky junction but not the p-GaN/AlGaN/GaN p-i-n junction. From Figure 8b, it can be concluded that the V_TH_ shift is inversely related to the I_GSS_.

## 4. Discussion

The mechanism of the BTI effect of the p-GaN gate HEMTs is analyzed as follows. As shown in Figure 9a, when the gate was initially forward-biased, the holes in the p-GaN were repulsed to the quantum well at the p-GaN/AlGaN interface. These holes electrically attract electrons to accumulate in the quantum well at the AlGaN/GaN interface. The channel is, thus, switched on. The V_TH_ depends on the capability of gate bias to incur hole accumulation at the p-GaN/AlGaN interface. During the BTI measurements, the fresh V_TH_ is usually very low because the holes in the p-GaN can be immediately repulsed to the quantum well at the p-GaN/AlGaN interface. After a certain time of BTI stress, these accumulated holes gradually emit from the p-GaN to the GaN channel over the AlGaN barrier. These lost holes are difficult to be supplemented from the gate terminal because of the metal/p-GaN Schottky barrier. The p-GaN layer, therefore, hosts more and more net negative charges. A higher gate bias is thus required to switch on the GaN channel, corresponding to a high V_TH_. In Figure 9b, during the TDGB stress, the metal/p-GaN Schottky junction is degraded, inducing a barrier lowering effect on this Schottky barrier. The holes from the gate metal are much easier to transport to the p-GaN layer, thus effectively suppressing the BTI effect. The trapping effect could also impact the BTI behavior which, however, should be a slow process that is different from the fast-charging effect in our measurement, as claimed by Yang et al. [15]. It is indeed difficult to unambiguously distinguish the root cause of the BTI effect because the trapping and charging effects usually induce the same impact. The more injected holes can also combine with the trapped electrons to suppress the BTI effect by a certain level. Stockman et al. pointed out that electron trapping by the AlGaN/GaN interface traps takes place when the gate bias is low, and the hole trapping by the Mg negative traps in the AlGaN barrier takes place when the gate bias is high [27]. Tang et al. [28] believed the electron trapping that induces the positive V_TH_ shift is dominated by the electron trapping at the AlGaN/GaN interface, while the negative V_TH_ shift is due to the generated light emission that optically pumps the electron traps, where the UV emission is generated by the recombination of the injected holes and 2DEG [29]. He et al. found that fast measurements can probe a monotonous increase of the V_TH_, while the slow DC measurement observed the same trend as Tang. They proposed that the electron trapping takes place at the AlGaN barrier and the depletion region of the p-GaN, so the trapped electrons are easy to be released, while the holes accumulated at the p-GaN/AlGaN interface are difficult to be released. Therefore, the fast measurement is quick enough to capture the trapping by the electrons [30].

It can be seen that in Figure 6b, after each TDGB stress, a long enough interval time was given to release the possible trapping effect, but the BTI behaviors of the fresh and stressed devices are still totally different. It is, therefore, at least plausible to conclude that the suppressed BTI effect is directly induced by the more injected holes. The root cause of the BTI effect itself is still an open topic in the future.

To verify our hypothesis, a technology computer-aided design (TCAD) simulation by Sentaurus was then conducted. The simulated p-GaN gate stack contains a 200 nm GaN channel, a 15 nm Al_0.2_Ga_0.8_N barrier layer, and a 70 nm p-GaN layer. The hole concentration in the p-GaN layer is set to be 3 × 10^17^ cm^−3^. As shown in Figure 10, the hole current was monitored during the simulation under the gate bias voltage of 10 V. The work function of the gate metal is tuned from 4.5 eV in Figure 10a to 5.2 eV in Figure 10b to mimic the potential barrier lowering effect, which means the Schottky barrier height in the latter case is 0.7 eV lower. We can clearly see that the hole current significantly improves, proving the rationality of our conclusion.

## 5. Conclusions

In conclusion, the mechanism of the BTI effect of the p-GaN gate HEMTs was investigated. By lowering the Schottky barrier height with the TDGB stress, the BTI effect was successfully suppressed. For the first time, it is experimentally proven that the hole loss in the p-GaN layer induces the BTI effect. The key to eliminating the BTI effect is to balance the hole injection and hole ejection from the p-GaN layer. However, the metal/p-GaN Schottky junction destroys this balance. Ohmic p-GaN gate contact is a solution which, however, suffers a low V_TH_. This work uncovers the cause of the BTI effect of p-GaN gate HEMTs and points out a direction for designing high-reliability HEMTs with a stable and applicable V_TH_.

## Figures and Tables

**Figure 1 micromachines-14-01042-f001:**
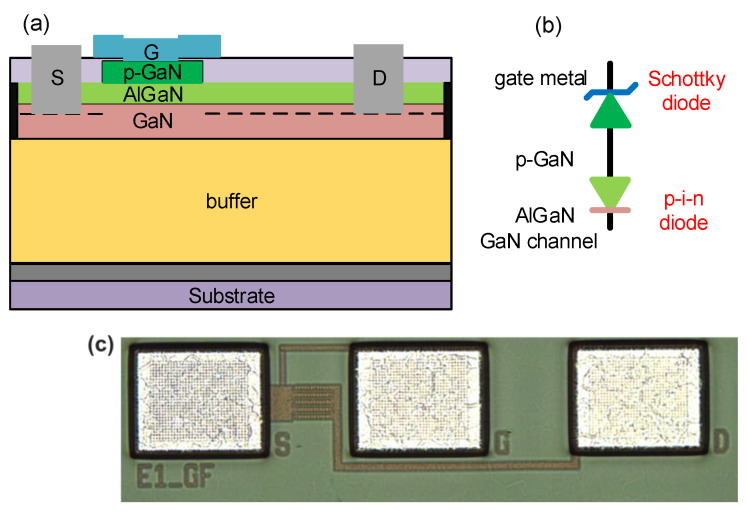
(**a**) The cross-sectional schematic structure of the p-GaN gate HEMTs. (**b**) Circuit model of the p-GaN gate structure, where the gate metal/p-GaN heterostructure forms the Schottky diode and the p-GaN/AlGaN/GaN heterostructure forms the p-i-n diode. (**c**) Micrograph of the measured device where the S, G, and D denoted source, gate, and drain separately.

**Figure 2 micromachines-14-01042-f002:**
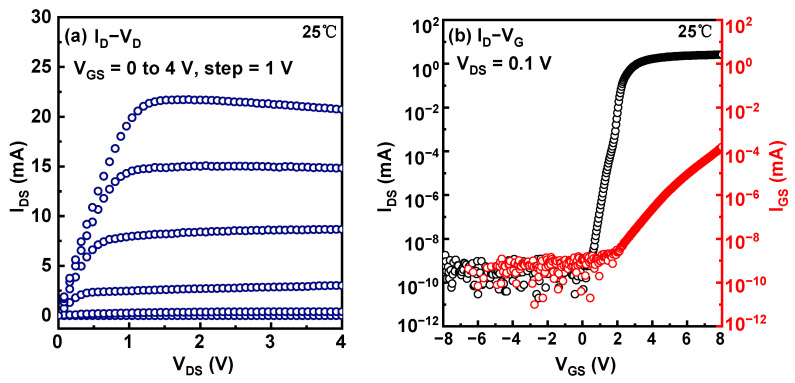
(**a**) Output (blue curve) and (**b**) transfer (black curve) characteristics, and the gate leakage curve (read curve) of the p-GaN gate HEMTs.

**Figure 3 micromachines-14-01042-f003:**
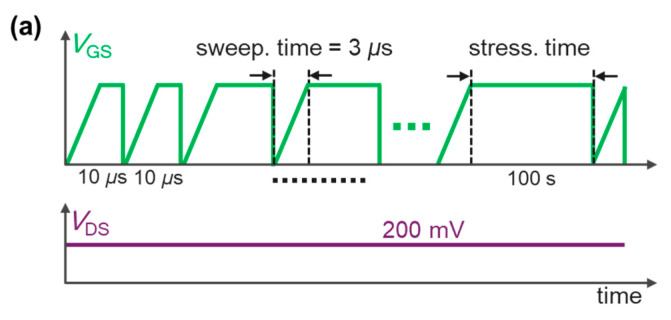
(**a**) BTI eMSM sequence during stress. (**b**) The fast I_D_-V_G_ sweep of the device during the BTI stress where the black arrow denotes the V_TH_ shift direction versus stress time. (**c**) V_TH_ versus stress time under different gate stress voltages.

**Figure 4 micromachines-14-01042-f004:**
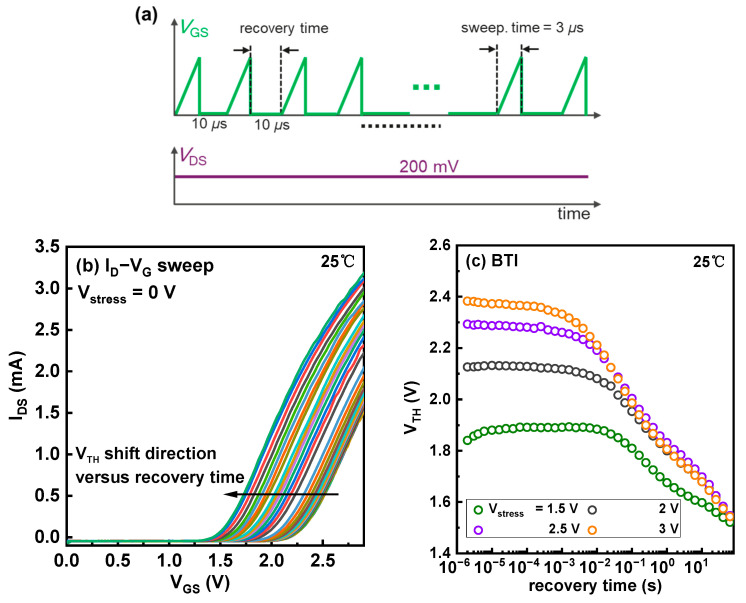
(**a**) BTI eMSM sequence during recovery. (**b**) The fast I_D_-V_G_ sweep of the device during BTI recovery where the black arrow denotes the V_TH_ shift direction versus recovery time. (**c**) The recovery of V_TH_ versus the recovery time.

**Figure 5 micromachines-14-01042-f005:**
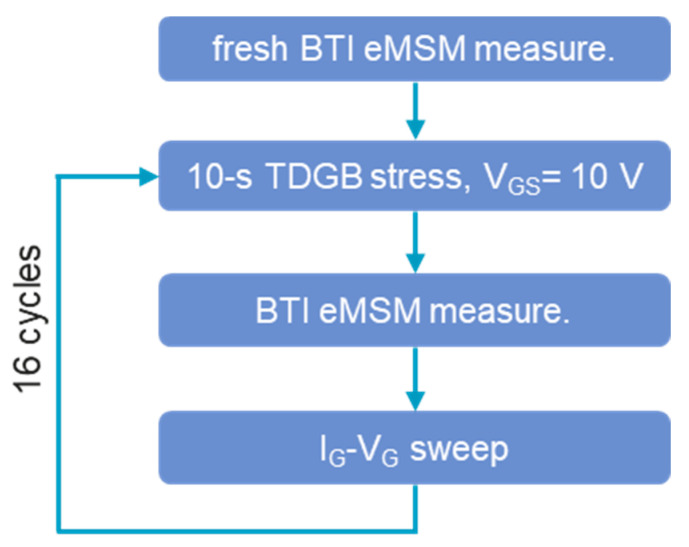
The experiment flow of suppressing the BTI effect by TDGB stress.

**Figure 6 micromachines-14-01042-f006:**
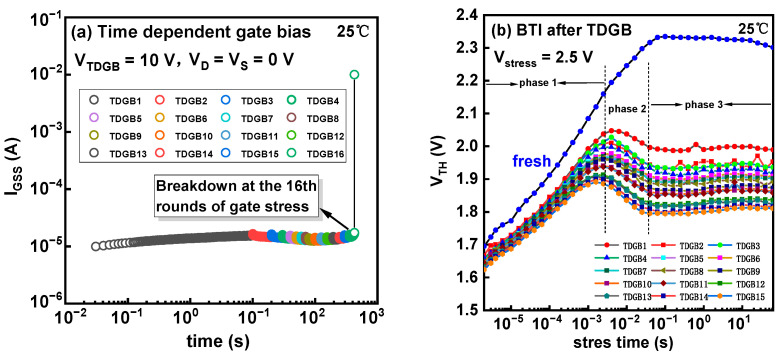
(**a**) I_GSS_ versus gate stress time. (**b**) The V_TH_ over BTI stress time after the TDGB stress.

**Figure 7 micromachines-14-01042-f007:**
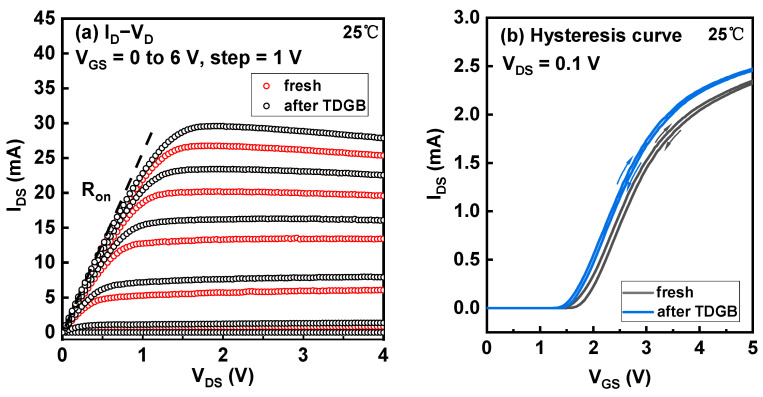
(**a**) The output and (**b**) hysteresis curves of the fresh and TDGB-stressed devices.

**Figure 8 micromachines-14-01042-f008:**
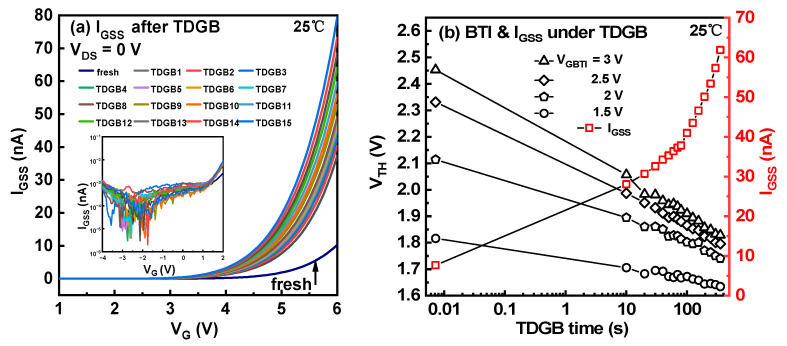
(**a**) The I_GSS_-V_G_ curves of the device under TDGB stress. (**b**) V_TH_ and I_GSS_ versus the stress time.

**Figure 9 micromachines-14-01042-f009:**
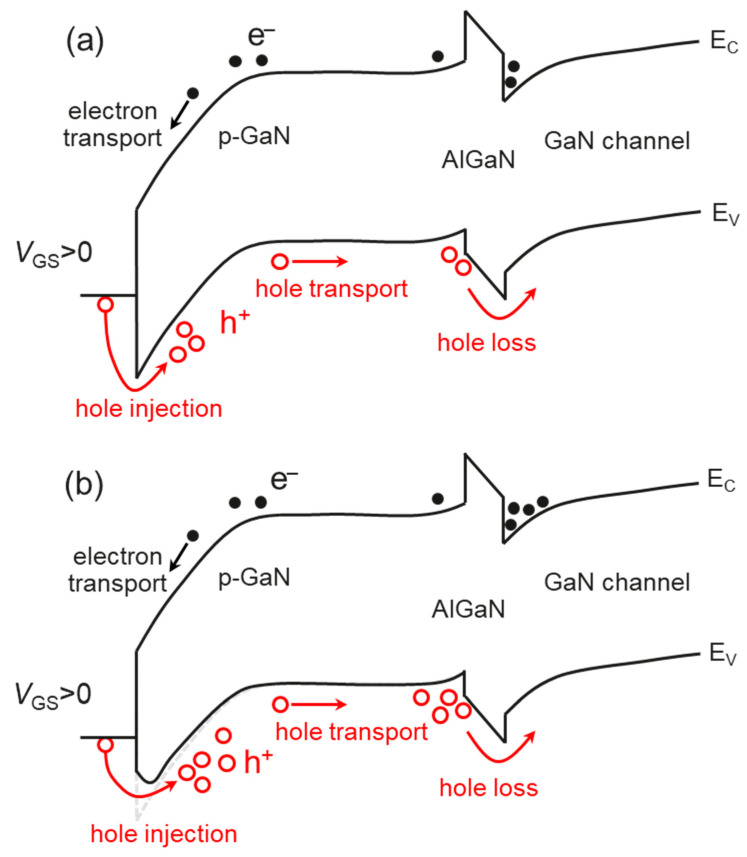
The schematic band diagram of the (**a**) fresh p-GaN gate and the (**b**) stressed p-GaN gate that has a lower Schottky potential barrier.

**Figure 10 micromachines-14-01042-f010:**
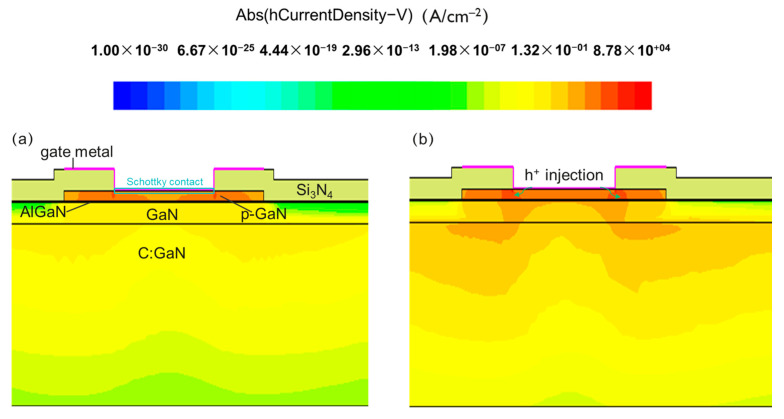
Simulated hole current density of the device with a (**a**) Schottky work function = 4.5 eV and (**b**) Schottky work function = 5.2 eV under 10 V gate bias.

## Data Availability

Data presented in this article are available on request from the corresponding author.

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
