# Peer review of "Revealing the Mechanism of the Bias Temperature Instability Effect of p-GaN Gate HEMTs by Time-Dependent Gate Breakdown Stress and Fast Sweeping Characterization"

_micromachines, 2023, doi:10.3390/mi14051042_

Round 1
Reviewer 1 Report
Review:
· Authors study threshold voltage shift of p-GaN gate HEMT under bias temperature instability by fast sweeping characterizations.
· They infer that TDGB stress can reduce instability of threshold voltage.
· Authors also provide a TCAD simulation to support their conclusion that Schottky barrier height lowering improves hole injection.
Recommendations:
· Micrograph of the studied device is required in Fig 1.
· The detail’s epitaxial structure of the device is necessary to conclude “For the first time, it is experimentally proven that the hole loss in the p-GaN layer induces the BTI effect”.
· Authors should provide full name of the measurement unit mentioned in lines 69-70 used and some details about its mechanism.
· A picture explaining the experimental setup is recommended.
· Adequate references should be provided supporting the text in lines 109-112.
· Correct the spelling of Schottky in line 125.
· In which plat form the TCAD simulation has been done?
Correct the spelling of Schottky in line 125
Reviewer 2 Report
This paper has discussed on mechanism of the bias temperature instability(BTI). The time-dependent gate breakdown stress (TDGB) with short time suppressed effectively BTI. It is interesting to the readers, but the discussion is not enough to explain the mechanism of BTI applied TDGB. I recommend the major revision according to the following comments. Especially please discuss the relationship between BTI and TDGB.
1. In this paper, the characteristics at only one temperature were measured in spite of paper about temperature instability. Could you add the reasons to measure at only room temperature?
2. Figure 5(b) is the most important for this paper. But there is no explanation including the mechanisms. Please add detail explanation in section 3.2 and/or 4.
3. line 168: There is less information about TCAD. Please add the simulation structure and used parameters.
4. line 172: Authors described that the Schottky barrier becomes low for Vg = 10 V. Please add the reasons why Schottky barrier becomes low.
5. line 141 What is HTGB?
6. Some figures aren't suitable for readability. Please revise following figures properly.
Figure 1(b) I don't understand where Schottky diode, p-GaN and pin diode. Please show by arrows.
Figure 5(b) There are no graph legends except for "fresh".
Figure 6(b) Please show hysteresis loop direction, whether clockwise or reverse.
Figure 9 Please show the names for each parts (p-GaN, GaN channel, Schottky and so on )
Reviewer 3 Report
In order to enhance the understanding by the unfamiliar reader, I'd suggest to extend the definition and description of bias temperature instability and how this is normally evaluated.
The experimental procedure could be better described. I suggest to extend it and position it in a separate subsection of Sec. 2. A flow chart of the test might also help.
In GaN, it is not easy to experimental assess damage or trapping mechanisms that could experience recovery in a relatively short time against those mechanisms that involve failure. For example, there are dynamic mechanisms with a wide range of time constants involved from very fast to very slow in the range of seconds and beyond, which also depend on the technology, see: Angelotti, et al. Trapping Dynamics in GaN HEMTs for Millimeter-Wave Applications: Measurement-Based Characterization and Technology Comparison. Electronics 2021, 10, 137. https://doi.org/10.3390/electronics10020137 Could the authors comment on this aspect and what kind of hypothesis did they applied in this sense?
In Sec. 3-1, what do the authors mean by "a semiology plot of these transfer curves"?
Round 2
Reviewer 1 Report
All questions were answered by the authors. Accept it as is, thank you.
Minor editing of English language required
Author Response
Thanks a lot.
Reviewer 2 Report
The authors reply all comments properly.
Author Response
Thanks a lot.
Reviewer 3 Report
Thanks for addressing the comments. However, I noticed from the first plot that the pulsed excitations are in the us-ms range, which is in the range of the time constants typically shown by de-trapping. At the same time, the authors later mention 50 ns regarding BTI effects.
I still believe that the observations shown might be influenced by trapping and the authors should explain more in detail how to account for it. Also, please enlarge citation list and literature comparison in this context.
Author Response
Thanks for the comment. The charging effect and trapping effect have the totally same electrical impact, which is therefore unable to be distinguished anyhow. We can only make conclusions by the observation and give a plausible conclusion. The trapping impact on the BTI effect itslft is not excluded in our clusion actually.
We have added the following part to the revised manuscript to include both the two possiblities.
The more injected holes can also combine with the trapped electrons, so that to suppress the BTI effect by a certain level. It can be seen that in Figure 6b, after each TDDB stress, long enough interval time was given to release the possible trapping effect, but the BTI behaviors of the fresh and stressed devices are still totally different. It is therefore at least plausible to conclude that the suppressed BTI effect is directly induced by the more injected holes. The root cause of the BTI effect itself is still an open topic in the future.
Round 3
Reviewer 3 Report
Thanks for this add-on, but I believe that the article would have been better with a larger contextualization and discussion of the impact of trapping and related techniques on this BTI characterization.
Please see if you can improve it by adding literature context in the final version.
Author Response
Thanks a lot for the comment. Related references and explanation about the trapping effect has been supplemented in the revised manuscript as follows.
Stockman et al. pointed out that electron trapping by the AlGaN/GaN interface traps takes place when the gate bias is low, and the hole trapping by the Mg negative traps in the AlGaN barrier takes place when the gate bias is high [27]. Tang et al. [28] believed the electron trapping that induces the positive VTH shift is dominated by the electron trapping at the AlGaN/GaN interface, while the negative VTH shift is due to the generated light emission that optically pumps the electron traps, where the UV emission is generated by the recombination of the injected holes and 2DEG [29]. He et al. found the fast measurements can probe a monotonous increase of the VTH while the slow DC measurement observed the same trend as Tang. They proposed that the electron trapping takes place at the AlGaN barrier and the depletion region of the p-GaN, so the trapped electrons are easy to be released, while the holes accumulated at the p-GaN/AlGaN interface are difficult to be released. Therefore, the fast measurement is quick enough to capture the trapping by the electrons [30].